# Meet challenges of RTT Jitter, A Hybrid Internet Congestion Control Algorithm

## ABSTRACT

Congestion control has been a fundamental research focus in web transmission for over 30 years. However, with diverse network scenarios like cellular networks and WiFi, traditional models might no longer accurately describe current network conditions – we empirically observe that the minimum round-trip time ($RTT_{min}$) still varies under different network conditions, challenging the assumption of its constancy in traditional models. In this paper, we model it as a normal distribution based on our measurements and propose a novel congestion control algorithm LingBo. LingBo consists of two phases: an offline trained decision model to achieve goals under different $RTT_{min}$ distributions, and an online perception scheme to detect the current $RTT_{min}$ distribution. We evaluate LingBo in various network environments and find it consistently performs well in terms of power metric and throughput compared to recent state-of-the-art baselines.

## CCS CONCEPTS

• Networks → Network control algorithms.

## KEYWORDS

Congestion Control, RTT Jitter, Imitation Learning

### ACM Reference Format:
Anonymous Author(s). 2018. Meet challenges of RTT Jitter, A Hybrid Internet Congestion Control Algorithm. In *Proceedings of Make sure to enter the correct conference title from your rights confirmation emai (Conference acronym 'XX)*. ACM, New York, NY, USA, 9 pages. https://doi.org/XXXXXXX.XXXXXXX

## 1 INTRODUCTION

Over the past three decades, congestion control (CC) has remained a research hotspot in the field of computer infrastructure. There are two main reasons for the continued attention to congestion control. Firstly, congestion control is a cornerstone algorithm of the modern internet and holds significant importance for various tasks conducted over networks. Secondly, network conditions have undergone significant changes due to the development of related technologies and increased application demands. Traditional wired networks continue to evolve, with a growing need for long-distance transmission, such as cross-border and intercontinental networks. Meanwhile, wireless network technologies have matured, offering

convenience to users through technologies like WiFi and cellular networks while also presenting new challenges in network transmission.

Queuing delay has served as a direct or indirect signal for many CC algorithms to perceive network conditions such as BBR [6], Copa [5], Aurora [14]. However, the methods used by these algorithms to calculate queuing delay is $RTT − RTT_{min}$. Implicit in this approach is the assumption that the $RTT_{min}$ is a relatively stable value. Through measurements spanning nearly 50 hours, we find that this assumption holds true in wired networks. However, in wireless scenarios (such as WiFi or cellular network), there are significant jitters in $RTT_{min}$, which poses challenges to the previous algorithms' network modeling and leads to poor performance in scenarios with high jitter.

Perceiving and handling RTT jitter is indeed a challenging task. Unlike the exploration of maximum bandwidth, exploring $RTT_{min}$ can potentially lead to a loss in bandwidth utilization. Balancing the accuracy of $RTT_{min}$ jitter perception and bandwidth utilization is a challenging design aspect. Furthermore, optimizing objectives in the presence of RTT jitter is also a significant challenge.

So, in our work, we collect nearly 50 hours RTT traces without queuing. Shifting away from the previous perspective of treating it as a fixed value, we consider the $RTT_{min}$ as a normal distribution. Then, we divide the algorithm into two modules: perception and decision. The perception module is responsible for sensing the network's $RTT_{min}$ distribution, while the decision module outputs the current congestion window (cwnd) based on the already established network model. In the offline training phase, we assume that the network's $RTT_{min}$ distribution is known. Within each time interval, we design target values using domain knowledge and train the model via imitation learning. In the online perception phase, we use expert knowledge to construct the perception frame and consider the decision model to determine the details such as the duration of the perception phase.

To demonstrate the algorithm's performance, we conduct experiments with 15 baselines covering both classical algorithms and recently published algorithms in both emulation and real-world environments. In the emulation environment, we validate the algorithm's robustness and compare its performance under different $RTT_{min}$ traces including three network scenarios: wired, WiFi, and cellular networks. Our algorithm achieves the highest power95 and improves with other algorithms from 18%-40x. Specifically, LingBo outperforms the BBR (the only one that achieves higher throughput) by 92% in average power95. In the real-world environment, we establish servers in four regions and conduct actual packet transmission experiments. LingBo achieves the highest throughput and the third highest power95. LingBo shows a 155%-6x improvement in throughput compared to Vivace [8] and Copa (top two algorithms in power95)

In general, we summarize the contributions as follows:

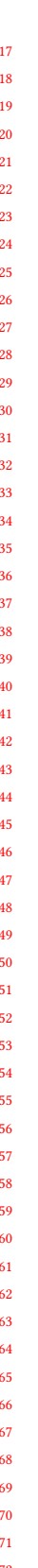

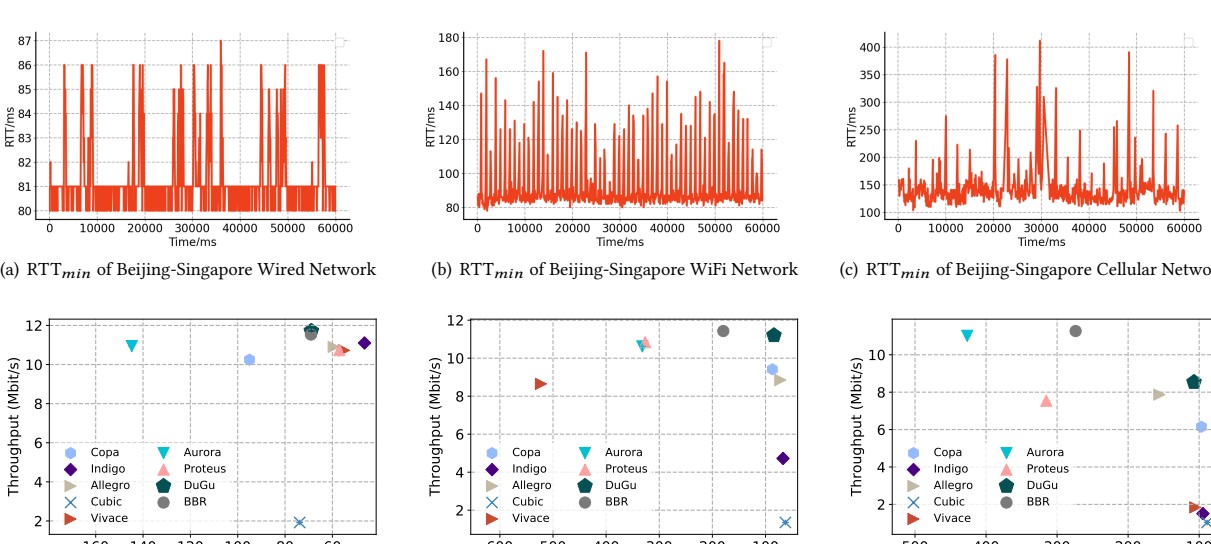

(a) $RTT_{min}$ of Beijing-Singapore Wired Network

(b) $RTT_{min}$ of Beijing-Singapore WiFi Network

(c) $RTT_{min}$ of Beijing-Singapore Cellular Network

(d) Performance in Beijing-Singapore Wired Trace

(e) Performance in Beijing-Singapore WiFi Trace

(f) Performance in Beijing-Singapore Cellular Trace

**Figure 1: $RTT_{min}$ Jitter and Algorithm Performance**

- We collect RTT traces without queuing for nearly 50 hours and conduct emulation experiments to illustrate the impact of $RTT_{min}$ jitter on algorithm performance. We model $RTT_{min}$ as a normal distribution rather than a fixed value. (§3)
- We propose a novel algorithm, LingBo, a hybrid congestion control algorithm divided into decision and online perception, to address the challenges posed by $RTT_{min}$ jitter. (§4)
- We conduct extensive evaluations in both emulation and real-world environments covering wired, WiFi, and cellular network scenarios with 15 baselines. LingBo consistently achieves competitive performance in throughput and power95. (§5)

## 2 RELATED WORK

Congestion control perceives the network state through signals such as packet loss and delay, and adjusts the size of the cwnd to control the data transmission rate. Some past research efforts focus on assessing network conditions by relying on individual network signals, such as packet loss [13] [11] and one-way delay [5]. Other research aims to create a comprehensive network model by considering multiple network signals in decision, such as BBR [6] and other learning-based methods such ass Orca [2] and Aurora [14]. These algorithms typically model the network via the current RTT substrates the minimum historical RTT as queuing delay, using this value to determine network congestion. However, this modeling approach overlooks the issue of RTT jitter [4]. Even in situations where there is no packet queuing, the RTT (commonly named $RTT_{min}$) can inherently exhibit fluctuations, which is particularly common in wireless networks (due to channel fluctuations, scheduling in uplink/downlink directions at BS, etc) [4] [1]

Previous research has also recognized these issues and proposed various solutions. Some research endeavors periodically probe the

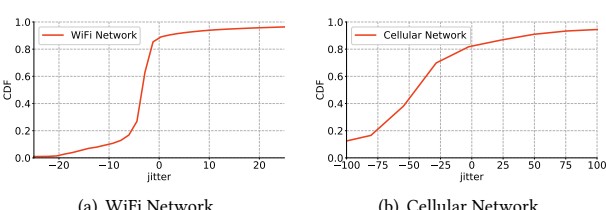

(a) WiFi Network

(b) Cellular Network

**Figure 2: The CDF of jitter**

$RTT_{min}$ such as BBR and Copa while others opt to introduce additional information, such as the router [10] [20] or physical layer information [21]. However, periodic probing may not effectively address the within-cycle jitter, and router or physical layer information is not accessible to end-to-end internet congestion control algorithms, as it requires modifications to the entire network link. Some algorithms utilize as conservative strategies as possible to counteract jitter, but inevitably result in a decrease in bandwidth utilization, such as C2TCP [1] and DeepCC [3].

## 3 $RTT_{min}$ JITTER IN NETWORK

### 3.1 $RTT_{min}$ Trace

$RTT_{min}$ jitter is widely present in various network environments, but there has been a lack of large-scale measurement results. To better illustrate the $RTT_{min}$ jitter, we fix the sender's sending cwnd as 2 [1] and sent packets from Beijing to clients located in Beijing, Hong Kong, Singapore, and Frankfurt. We collect $RTT_{min}$ traces for nearly 50 hours, encompassing three network types: wired,

---

[1]This is a commonly used method for estimating $RTT_{min}$ such as [6] [5].

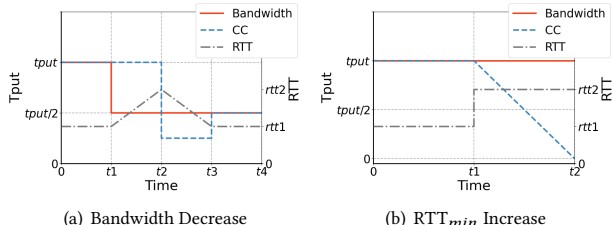

(a) Bandwidth Decrease      (b) $RTT_{min}$ Increase

**Figure 3: The Difference between $RTT_{min}$ Jitter and Bandwidth Variation**

WiFi, and cellular network, and including two states: stationary and mobile.

In Figure 1(a) 1(b) 1(c), we plot representative $RTT_{min}$ traces under three different network conditions and plot the cumulative distribution function of $RTT_{min}$ jitters in WiFi and cellular networks as Figure 2. We can see that jitter is widely present, particularly in wireless network scenarios. The $RTT_{min}$ in the wired network remains relatively stable, with jitter at a few milliseconds. In the WiFi network, the jitter is around several tens of milliseconds, while in the mobile network, the jitter is even greater.

## 3.2 Impact of $RTT_{min}$ Jitter on CC

To visually demonstrate the impact of $RTT_{min}$ jitter on congestion control algorithms, we conduct emulation experiments using Mahimahi [16]. We fix the bandwidth as 12mbps and use different $RTT_{min}$ trace to test different algorithms including BBR, Cubic, Copa, Indigo [22], Allegro [7], Vivace [8], Proteus [15], Aurora [14], Dugu [12] as Figure 1(d) 1(e) 1(f). We can see most algorithms perform well in the wired trace, but in the WiFi trace, most algorithms fail to achieve the balance between high throughput and low latency. In the cellular trace, all algorithms either have low bandwidth utilization or excessively high latency.

To further uncover the reasons behind the performance degradation caused by $RTT_{min}$ jitter, we conduct a simple analysis and plot Figure 3, comparing the effects of bandwidth decrease and $RTT_{min}$ increase. Figure 3(a) demonstrates the changes in RTT and the CC algorithm caused by a decrease in bandwidth. When the bandwidth decreases at time $t_1$, the RTT increases due to the increased queuing delay. Upon perceiving this RTT signal at time $t_2$, the CC algorithm reduces the sending window to clear the queue and then restores the sending window to match the current bandwidth at time $t_3$. However, if the change in RTT is not caused by the queuing delay but by $RTT_{min}$, the situation would be different. As shown in Figure 3(b), when $RTT_{min}$ increases, the CC algorithm may misjudge the current network condition and decrease the sending rate in the hope of restoring the RTT to its previous value. This can result in a continuous decline in the CC sending rate.

## 3.3 Model for $RTT_{min}$ Jitter

In the past, $RTT_{min}$ was typically considered as a single value, perceived by periodically updating it. However, based on the results of our measurements, we find due to the presence of jitter, a single value is insufficient to model $RTT_{min}$ accurately. We plot probability density histograms as Figure 4 to illustrate the jitter in

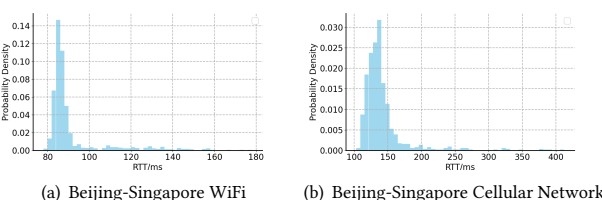

(a) Beijing-Singapore WiFi      (b) Beijing-Singapore Cellular Network

**Figure 4: $RTT_{min}$ Histogram**

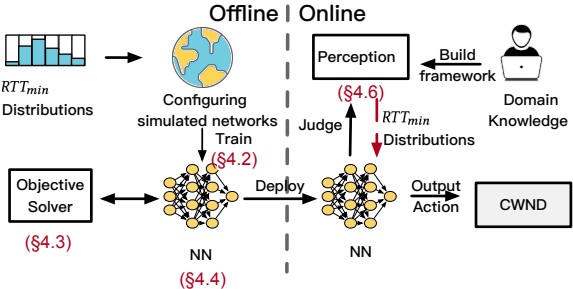

**Figure 5: LingBo System Overview**

$RTT_{min}$ for both WiFi and cellular networks (shown in Figure 1(b) and Figure 1(c)). The probability density of $RTT_{min}$ exhibits a clear peak, with the probability density function gradually decaying on both sides. As the distance from the mean increases, the probability density gradually decreases. Based on this observation, we modify the previous approach of considering $RTT_{min}$ as a single value and instead treat it as a normally distributed variable.

## 4 LINGBO

### 4.1 LingBo Overview

As depicted in Figure 5, we employ imitation learning during the offline phase to train a decision model with given network parameters as the decision module. Then, during the online phase, the perception module obtains the network parameters and provides the $RTT_{min}$ distribution to the decision model, which outputs the final cwnd.

**Decision Module** Due to the challenges posed by $RTT_{min}$ jitter, it is difficult to address them effectively using manually designed rules. Therefore, we aim to employ a learning-based approach to automatically derive strategies that can adapt to various network conditions. To avoid the challenges associated with designing reward functions in reinforcement learning, we adopt imitation learning as the training algorithm, learning from the designed objectives. We introduce our simulation environment design (Sec 4.2), objective design and calculation (Sec 4.3), neural network architecture (Sec 4.4), as well as the training methodology (Sec 4.5).

**perception module** (Sec 4.6) Perception is a challenging task as we aim to perceive the distribution of $RTT_{min}$ as accurately as possible without compromising the algorithm's performance. We combine domain knowledge with the pre-trained decision model to jointly determine the perception phase.

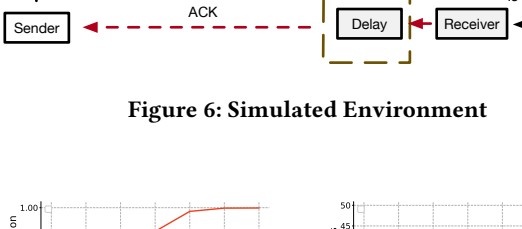

**Figure 6: Simulated Environment**

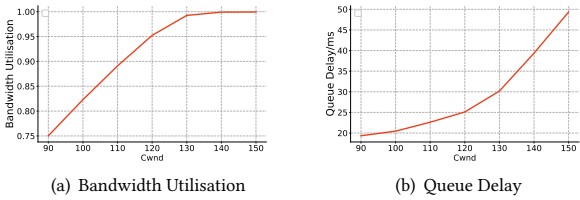

(a) Bandwidth Utilisation

(b) Queue Delay

**Figure 7: Bandwidth Utilisation and Queue Delay with Different Objectives**

## 4.2 Simulated Environment

Inspired by previous work [16], we design a faithful simulation environment as depicted in Figure 6. When the sender receives an ACK packet sent back from the receiver, it utilizes the information obtained from this feedback to update its state. Every 100ms, the neural network makes a decision and updates the new congestion window size. Data packets are transmitted from the sender and enter the network part based on the cwnd. Building upon the previous simulator, we model the network with three components: packet loss part, delay part, and link bandwidth part. In the packet loss part, we simulate random packet loss in the network using a probability ranging from 0 to 1. In the delay part, we simulate the entry of packets into the network link and the time taken for acknowledgments to be sent back from the receiver using randomly generated values based on a given distribution for the $RTT_{min}$. In the link bandwidth part, we utilize real-world bandwidth traces to simulate the availability of the network for sending packets.

## 4.3 Objective Design and Calculate

Due to the presence of $RTT_{min}$ jitter, designing the objectives for imitation learning becomes a new challenge. Traditional metrics such as the Bandwidth-Delay Product (BDP) may not be suitable for scenarios with jitter. To illustrate this issue more intuitively, we design a simple experiment using our simulator lasting 200s. We model $RTT_{min}$ as a normal distribution with a mean of 100ms and a variance of 10ms, and the bandwidth is set to 1 packet per millisecond. We compare the bandwidth utilization and queuing delay under different cwnd as Figure 7. We can observe that when setting the cwnd to the BDP (i.e., 100), we only achieve 82% bandwidth utilization and still experience approximately 20ms of queuing delay. This is because, when entering the delay part, if the $RTT_{min}$ of the previous packet increases, the subsequent packets have to wait in the queue until the previous packet is sent. This results in lower

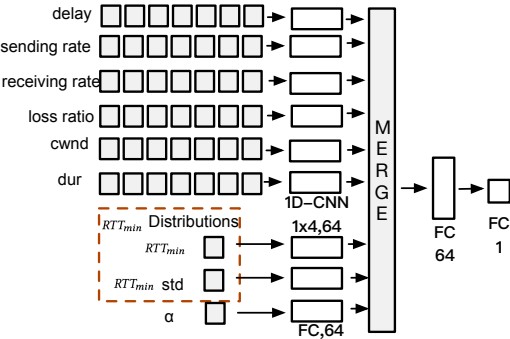

**Figure 8: NN Architecture**

bandwidth utilization and larger transmission delays, even though the cwnd is the same as the BDP. As shown in Figure 7, it can be observed that to achieve higher bandwidth utilization, properly occupying the buffer queue is a way to mitigate jitter. Therefore, setting cwnd to 120 or 130 in this experiment, which corresponds to setting the target queuing delay as 2std or 3std (As we model $RTT_{min}$ as a normal distribution, 2 std can absorb 95% of the jitter, while 3std can absorb 99.7% of the jitter), is a more suitable choice to achieve the balance between low latency and high throughput.

When calculating the objective value at time $t$, we first make a copy of the current network state as a virtual network and compute the target value within this virtual network. We then simulate emptying all the data packets in the network up to $t'$, which represents the time at which the first packet sent by the sender after time $t$ arrives at the receiver. Then we compute how many packets $p$ should be sent from $t'$ to $t' + dur$ and occupy the buffer queue $tput \cdot \alpha \cdot std$. So the cwnd $c_t$ at time $t$ can be calculated as

$$
\begin{aligned}
c_t &= \frac{\int_{t'}^{t'+dur} p_x dx + \frac{\int_{t'}^{t'+dur} p_x dx}{dur} \alpha \cdot std}{dur} RTT_{min} \\
&= (1 + \frac{\alpha \cdot std}{dur}) \frac{RTT_{min}}{dur} \int_{t'}^{t'+dur} p_x dx \\
\alpha &\in [0, 3]
\end{aligned}
\tag{1}
$$

## 4.4 Neural Network Architecture

**input** At each decision time $t$, to better represent the current state of the network, we choose several network signals as state $s_t$.

$$
s_t = (d_t, r_t^s, r_t^r, l_t, c, dur_t, RTT_{min}, RTT_{min}\ std, \alpha) \tag{2}
$$

Here, $d_t$ is the estimated queuing delay, which is calculated by $RTT - RTT_{min}$. $r_t^s$ and $r_t^r$ are the sending rate and receiving rate respectively, which are calculated by the ACK. We use the EWMA method to smooth these three signals. $l_t$ is the loss ratio, $c_t$ is cwnd and $dur_t$ is the duration since the last action. Additionally, we employ the mean $RTT_{min}$ and standard deviation $RTT_{min}\ std$ to model the $RTT_{min}$ distribution. And we use $\alpha$ in Equation 1 to represent the selection of objective values. We design a history length of 8 for the first six state variables to balance the trade-off between information and computational cost and the last three states, which do not change over time, have an input length of 1.

**Table 1: Range of env. during the training.**

| Traces | mean of $RTT_{min}$ | std of $RTT_{min}$ | Loss | Buffer Size | $\alpha$ |
|---|---|---|---|---|---|
| 0.1-300Mbps | 1-400ms | 0%-30% mean | 0-30% | 1-2000 pkts | 0-3 |

**output** After observing state $s_t$, the agent outputs $a_t \in (-1, 1)$, which is then used to modify the cwnd $c_t$ to determine the value of $c_{t+1}$ for the subsequent time slot $t + 1$, following the equation $c_{t+1} = c_t \cdot (1 + a_t)$.

**Architecture** As shown in Figure 8, we use six 1D-CNN layers with channels=64 to extract the feature from $(d_t, r_t^s, r_t^r, l_t, c, dur_t)$. We utilize there 64-dim fully connected layers to extract the characteristics separately for $(RTT_{min}, RTT_{min}std, \alpha)$. Then we merge these vectors and feed them to a 64-dim full connected layer and a 1-dim full connected layer. Finally, The NN outputs a single scalar using $tanh$ activation function.

## 4.5 Training Methodology

**Environment setting** We use over 2000 real-world network traces from various scenarios such as wired, WiFi, and cellular networks in a total of 60 hours from Orca [2], DeepCC [3] FCC [17] and HSDPA [18]. Although we have collected a large amount of RTT traces, in order to construct a more comprehensive training environment, we use a method based on manual construction to generate the RTT settings for the training environment. In addition, we have set different network parameters such as packet loss rate and queue buffer size. The specific range of settings is shown in the Table 1.

**loss function** We use imitation learning [19] to minimize the distance between the current policy $\pi(\theta)$ and the expert policy $\pi^*$. For the current state $s$, we can update the model by minimizing the gap between the output $a$ of the current policy $\pi(s, \theta)$ and the expert action $\pi^*(s)$. So the squared-loss function of LingBo can be described as Eq. 3.

$$L_{\text{LingBo}} = \frac{1}{4}(a - \pi^*(s))^2 \tag{3}$$

## 4.6 Online Perception

Perceiving $RTT_{min}$ accurately is a challenging task, and we aim to leverage domain knowledge and the pre-trained decision model to estimate the $RTT_{min}$ distribution as accurately as possible without impacting bandwidth utilization.

**Domain Knowledge** Taking inspiration from the design principles of previous heuristic algorithms [6] [5], we divide the perception process into two parts: initial perception and periodic perception.

During the initial perception phase, we set cwnd as 2 at the beginning and we assume that there is no queuing of received packets in this state. Additionally, if there is no exploration of $RTT_{min}$ within a 10-second interval, the CC algorithm will decrease the cwnd and initiate a new round of $RTT_{min}$ probing.

**Pre-trained Decision Model** Although the previous heuristic algorithm provides a framework for our perception module, some issues still need to be addressed, such as determining when the perception phase ends and how much to decrease the cwnd during re-probing.

To address these two issues, we leverage decision models that have been trained offline. Before the first decision, we continuously update the perceived $RTT_{min}$ distribution in real-time and provide this information to the pre-trained model for decision. As the goal of the decision model is to get cwnd as $((1 + \frac{\alpha \cdot std}{dur}) \cdot BDP)$, so long as the $a_t \geq \frac{\alpha \cdot std}{dur}$, we consider that no queuing has occurred in the past. We store the currently obtained RTT measurements after the first decision in a temporary queue. If the action meets the above condition, we add these measurements to the $RTT_{min}$ queue and update the $RTT_{min}$ distribution, otherwise, the perception phase ends. When reentering the perception phase, we also leverage the knowledge of the decision model. We aim to set the cwnd to $\frac{BDP}{2}$ at this point, while the optimization objective of our model is Eq 1. Therefore, we set the cwnd as:

$$c_{t+1} = c_t \cdot (dur/(2 \cdot dur + 2 \cdot \alpha \cdot std)) \tag{4}$$

and restart the perception phase. In summary, the code for LingBo during online perception is as follows:

---
**Algorithm 1** LingBo online perception

---
1: Initialization: Decision Model $NN$, state $s$, queue of $RTT_{min}$ $q_{min}$, temporary queue $q_{tmp}$, time $t$, cwnd $c$, update duration $dur$, target queue delay $\alpha \cdot std$
2: **while** perception phase not ends **do**
3:     **while** receiving ACK **do**
4:         $RTT = get\_RTT(ACK)$
5:         **if** Before first decision **then**
6:             $q_{min} \leftarrow q_{min} \cup RTT$
7:         **else**
8:             $q_{tmp} \leftarrow q_{tmp} \cup RTT$
9:         **end if**
10:     **end while**
11:     **if** meet time to decision **then**
12:         $s \leftarrow update\_state(q_{min})$
13:         $a_t = NN(s)$
14:         **if** $a_t \geq \frac{\alpha \cdot std}{dur}$ **then**
15:             $q_{min} \leftarrow q_{min} \cup q_{tmp}$
16:             $q_{tmp} \leftarrow []$
17:             $s \leftarrow update\_state(q_{min})$
18:             $a_t = NN(s)$
19:         **else**
20:             perception phase ends
21:         **end if**
22:         $c_{t+1} = c_t \cdot (1 + a_t)$
23:     **end if**
24: **end while**

---

## 5 EVALUATION

To demonstrate the performance of LingBo [2], we compare it with 15 other algorithms, including classical and recently published approaches. We first test the robustness of LingBo under different network parameters, such as varying packet loss ratios and buffer sizes (Sec 5.1). Using the collected RTT traces, we conduct trace-driven emulation experiments on both fixed and variable bandwidth

---
[2]During the evaluation, we set the $\alpha$ in Eq 1 to 3 as discussed in Sec 4.3.

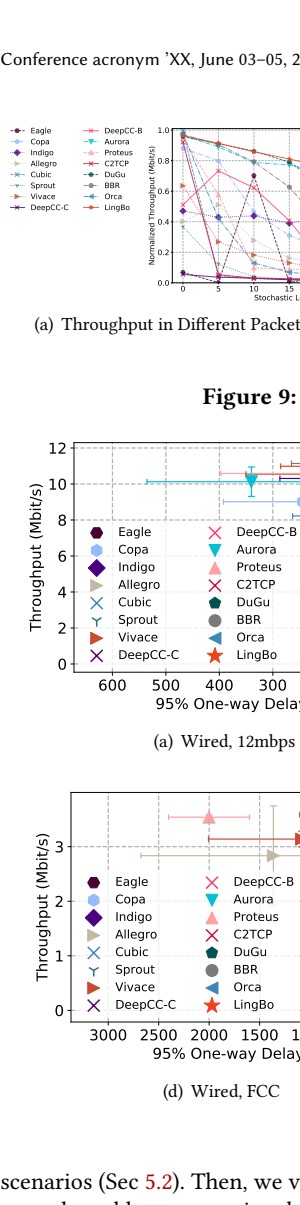

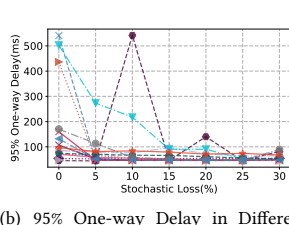

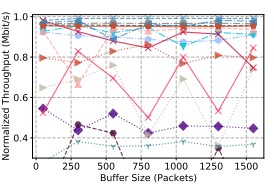

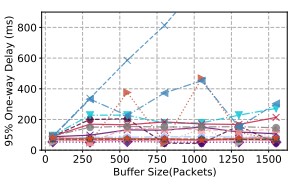

(a) Throughput in Different Packet Loss Ratios

(b) 95% One-way Delay in Different Packet Loss Ratios

(c) Throughput in Different Buffer Sizes

(d) 95% One-way Delay in Different Buffer Sizes

**Figure 9: The Robustness of `LingBo` under Different Packet Loss Ratios and Buffer Sizes**

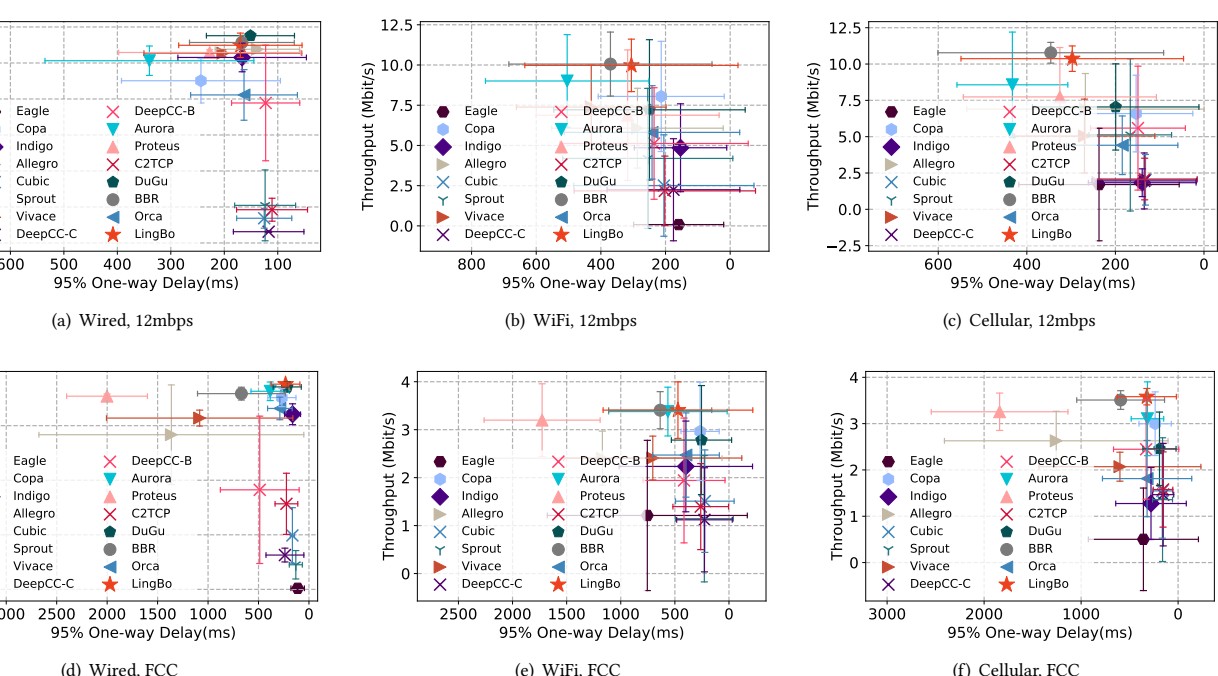

(a) Wired, 12mbps

(b) WiFi, 12mbps

(c) Cellular, 12mbps

(d) Wired, FCC

(e) WiFi, FCC

(f) Cellular, FCC

**Figure 10: The Results in Trace-driven Emulation**

scenarios (Sec 5.2). Then, we validate the performance of `LingBo` on real-world encompassing domestic, international, and intercontinental networks (Sec 5.3). In the end, we discuss `LingBo`'s fairness and TCP friendliness as Sec 5.4.

**Baselines** We take 15 state-of-the-art CC algorithms published in recent years as the baselines.**1) Cubic** [11]: a loss-based approach; **2) BBRv2(BBR)** [6], a model-based approach which detects maximum bandwidth and minimum RTT periodically, **3) Copa** [5]: a delay-based approach **4) Sprout** [23]: an approach addresses the uncertainty of cellular link variations and determines the network condition based on the observed packet arrival times at the receiver. **5) C2TCP** [1]: a delay-based approach specifically designed for cellular networks selects the minimum RTT within a certain period as $RTT_{min}$ **6) Auraro** [14]: a reinforcement-learning-based approach.**7) Indigo** [22]: an imitation-learning-based approach **8) Eagle** [9]: a reinforcement-learning algorithm that designs its reward function by imitating BBR. **9)-10) DeepCC** [3]: a hybrid algorithm specifically designed for cellular networks that utilize reinforcement learning to constrain the cwnd of heuristic algorithms and we

consider two variants using BBR **(DeepCC-B)** and Cubic **(DeepCC-C) 11) Orca** [2]: A hybrid algorithm that combines reinforcement learning with the Cubic. **12) DuGu** [12]: an imitation-learning-based approach with omniscient-like network emulator **13) PCC Allegro (Allegro)** [7], **14) PCC Vivace(Vivace)** [8], and **15) PCC Proteus (Proteus)** [15]: online-learning-based approaches with different optimization targets.

## 5.1 Robustness Analysis

As Figure 9, we conduct robustness analysis to test the performance in throughput and latency under different packet loss scenarios and various queue buffer sizes. We fix the bandwidth as 12mbps and consider the WiFi trace as shown in Figure 1(b) as $RTT_{min}$ trace. During the loss ratio test, we fix the queue buffer sizes as 500 packets and change the loss ratio from 0% to 30%. As shown in Figure 9(a) and Figure 9(b), `LingBo` is the only algorithm that performs well even with a 30% loss ratio. For different queue buffer sizes with 0% loss ratio, `LingBo` consistently maintains high throughput and low latency from 50-1800 packets buffer sizes as Figure 9(c) and Figure 9(d).

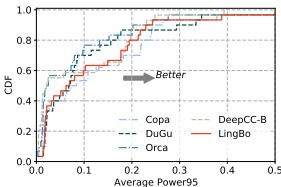

(a) Power95 in 12mbps Trace

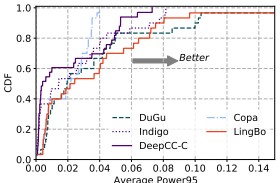

(b) Power95 in FCC Trace

**Figure 11: The CDF in Power95**

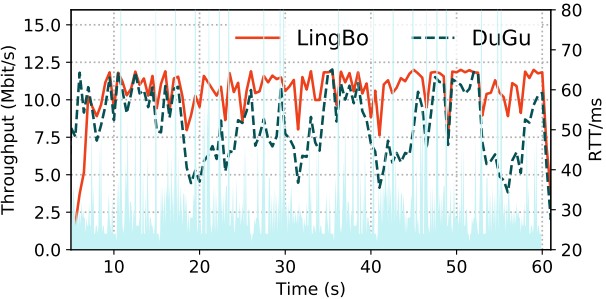

**Figure 12: Comparison of Whether to Model Jitter**

## 5.2 Trace-driven Emulation

To evaluate LingBo's performance in different network bandwidths and RTT settings, we conduct extensive emulation experiments based on real traces. We use Mahimahi [16] as the emulation tool and use fixed 12mbps trace and real-world bandwidth trace from FCC to emulate bandwidth. To show the performance in different $RTT_{min}$ scenarios, we use beyond 30 minutes $RTT_{min}$ trace including wired, WiFi, and cellular networks. We plot the 95% one-way delay and throughput as Figure 10.

**Wired Network** No matter whether in fixed bandwidth or varying bandwidth scenarios, most algorithms perform well. DuGu, BBR, Allegro, LingBo, and Indigo demonstrate excellent performance in terms of throughput and latency at 12mbps. Similarly, LingBo, DuGu, Copa, Auraro, and Orca exhibit outstanding performance in both throughput and latency under the FCC conditions.

**WiFi Network** The $RTT_{min}$ jitter in WiFi networks is much more complex compared to wired networks. LingBo consistently achieves a Pareto-optimal position in different scenarios, surpassing other algorithms. LingBo achieves nearly the highest throughput and, compared to BBR with similar throughput, reduces the average 95th percentile latency by 22%. Additionally, compared to Copa with a lower average 95th percentile latency, LingBo improves throughput by 21%.

**Cellular Network** The $RTT_{min}$ jitter in cellular networks can be the most complex, as the $RTT_{min}$ trace encompasses 4G and 5G networks and includes both mobile and stationary scenarios. In 12mbps, only LingBo, BBR, and Auraro achieve over 70% bandwidth utilization rate. Among them, LingBo reduced latency by 14%-31%. In FCC bandwidth trace, LingBo simultaneously achieves the highest throughput and lower latency, positioning itself in the top right corner of Figure 10(f).

**Power95** To visually demonstrate the superiority of LingBo compared to others, we choose a version of Kleinrock's power metric [22] named powed95 as the evaluation metric, which is calculated by:

$$power95 = \frac{throughput}{95\%delay} \tag{5}$$

We select the top 5 algorithms based on power95 ranking in fixed bandwidth and varying bandwidth scenarios respectively, and plot their CDF as Figure 11. LingBo, DuGu, and Copa perform well in both scenarios, and LingBo achieves the highest power95 in total. In Figure 11(a), LingBo, Copa, and DeepCC-B exhibit similar performance in most scenarios, while LingBo stands out with significantly higher power compared to other algorithms in most scenarios of FCC trace as Figure 11(b).

**Case Study: Whether to Model Jitter** To demonstrate the effectiveness of LingBo's modeling of $RTT_{min}$ jitter, we conduct a case study comparing it with another algorithm, DuGu, which is also based on imitation learning but handles jitter in a simpler manner. As Figure 12, we fix the bandwidth as 12mbps and use a cellular $RTT_{min}$ trace as the shaded area. It can be observed that while DuGu is able to handle small degrees of $RTT_{min}$ jitter, it becomes confused when there are rapid fluctuations in $RTT_{min}$ (such as 15s-18s, 41s-48s). DuGu struggles to differentiate whether the current increase in RTT is due to bandwidth reduction or $RTT_{min}$ jitter, leading to a decision to decrease throughput. In contrast, LingBo exhibits more stable throughput performance in the presence of such $RTT_{min}$ jitter.

## 5.3 Real-world Evaluation

To test the performance of LingBo in real-world scenarios, we deploy four servers globally, including a local network within a city (Beijing-Beijing), a domestic network (Beijing-Hongkong), a continental network (Beijing-Singapore), and an intercontinental network (Beijing-Frankfurt). We place the sender server in Beijing and modify the network type of the sender server to three types: wired, WiFi, and cellular networks. We run each algorithm 180s and repeat 5 times using random order on the Pantheon platform [22].

**Wired Network** As shown from Figure 13(a) to Figure 13(d), LingBo consistently achieves competitive performance with high throughput and low latency, outperforming existing algorithms notably in the Beijing-Frankfurt scenario. This demonstrates that in real-world wired scenarios with relatively low jitter, LingBo remains one of the most competitive algorithms.

**WiFi Network** Most algorithms do not perform as expected in WiFi scenarios, unlike in wired networks where most algorithms concentrate around the Pareto frontier. Some algorithms prioritize high bandwidth but result in significant queuing delay, while others focus on low latency but have low bandwidth utilization. LingBo achieves a good balance between throughput and delay, obtaining the second-highest throughput and the third-highest power95 ranking. Compared to Auraro with similar throughput, LingBo reduces average latency by 28%. In comparison to Copa and Vivace, which have higher power rankings, LingBo improves bandwidth by 23x and 142%, respectively.

**Cellular Network** In cellular networks, which present the most complex conditions, LingBo consistently achieves the best results in most tests, as clearly seen in Figure 13(i), 13(j), and 13(l). LingBo

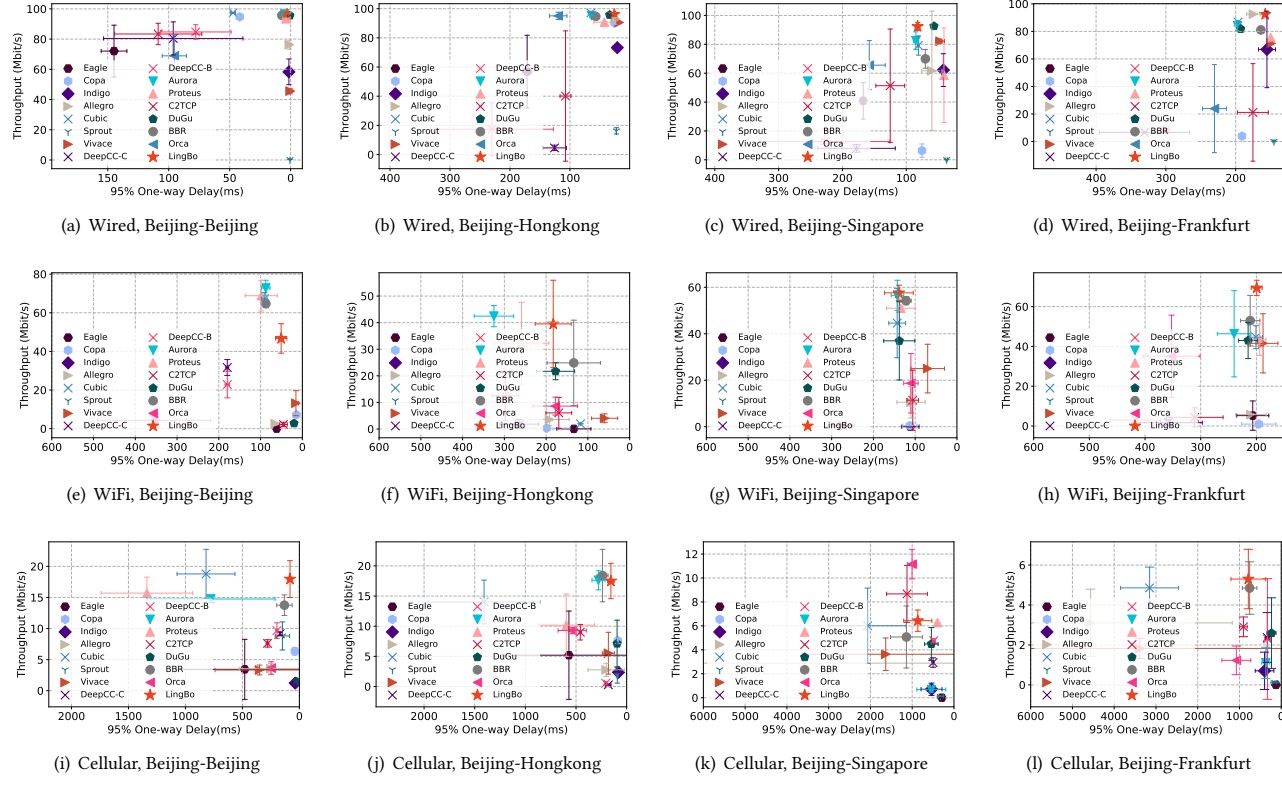

(a) Wired, Beijing-Beijing  (b) Wired, Beijing-Hongkong  (c) Wired, Beijing-Singapore  (d) Wired, Beijing-Frankfurt

(e) WiFi, Beijing-Beijing  (f) WiFi, Beijing-Hongkong  (g) WiFi, Beijing-Singapore  (h) WiFi, Beijing-Frankfurt

(i) Cellular, Beijing-Beijing  (j) Cellular, Beijing-Hongkong  (k) Cellular, Beijing-Singapore  (l) Cellular, Beijing-Frankfurt

**Figure 13: The Results in Real-world Evaluation**

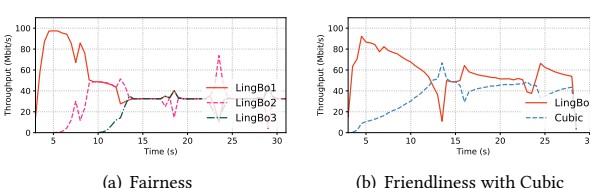

(a) Fairness  (b) Friendliness with Cubic

**Figure 14: The Fairness and Friendliness**

achieves the highest throughput, improving by 6% to 9x compared to other algorithms. Compared to Proteus and BBR, which have smaller throughput gaps, LingBo reduces queuing delays by 30% and 17% respectively. LingBo achieves the second-highest power95, surpassing Copa's (the highest power95) throughput by 50%.

## 5.4 Fairness and TCP Friendliness

The previous experiments have shown LingBo's great performance. However, fairness and TCP Friendliness are also very important as a CC algorithm.

**Fairness** LingBo's probing follows the frameworks of BBR and Copa, regularly updating the perceived $RTT_{min}$ parameter, ensuring fairness in the perception aspect. Furthermore, each LingBo flow shares the same optimization goal, guaranteeing fairness in the decision process. We conduct tests in real-world, where we send out 3 flows every 5 seconds. LingBo effectively distributes bandwidth evenly among the flows as Figure 14(a), demonstrating its capability to achieve fair bandwidth allocation.

**Friendliness** To test whether LingBo is too aggressive, we follow a similar approach to previous work [2] by selecting Cubic as the base TCP, which is the default TCP in most of today's Operating Systems. Cubic uses packet loss as a congestion signal, while our optimization target is before packet loss occurs, making us more conservative than Cubic (as observed in previous experiments such as Figure 13(i) 13(l)). We first send out a LingBo flow and then subsequently add a Cubic flow. We find that these two algorithms can coexist harmoniously as Figure 14(b).

**Remark** LingBo demonstrates excellent robustness, fairness, and TCP-friendliness, and achieves impressive performance. In emulation environments, it achieves the highest power95 and the second-highest throughput. In real-world scenarios, it stands as one of the most competitive algorithms for wired networks and attains the highest throughput and the third-highest power95 in wireless scenarios, showcasing a significant advantage over other algorithms.

## 6 CONCLUSION

To better address the jitter problem in congestion control, we conduct measurements for nearly 50 hours and model the $RTT_{min}$ as a normal distribution. We propose a new CC algorithm, LingBo, which combines a decision module via imitation learning and an online perception module. Through extensive experiments with 15 baselines in emulation and real-world scenarios, LingBo demonstrates competitive performances in both throughput and power95.

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
