# OpenReview forum: "Meet challenges of RTT Jitter, A Hybrid Internet Congestion Control Algorithm"
_ACM.org/TheWebConf/2024/Conference — TheWebConf24_

### Official Review · Reviewer_RmXB · 2023-11-17

**Novelty:** 5
**Technical Quality:** 5

**Review:**

This paper first demonstrate the RTTmin is not a stable value and may fluctuate heavily especially in wireless network by real network traces. Based on the observation, this paper further proposes a new TCP mechanism LingBo, which use an offline and online method to predict the RTTmin distributions and determine how to perceive the RTTmin. Based on the accurate RTTmin, LingBo can achieve better performance compared with other TCP mechanisms.

Strengths：

-	Although the observation for RTTmin is not new, it is the first one to prove it by real network traces. And it further tries to improve the perception phase of RTTmin.

-	The evaluation is solid and compared most of the relative TCP mechanisms.

Major weaknesses:

-	There are lack of demonstration for the effectiveness of RTTmin perception in the evaluation.

**Questions:**

It is a solid work. The observation is valuable and the solution is effective according the evaluation. Here are some comment which can perhaps further to discuss.

1)	This paper assume that RTTmin in wireless networks follows a normal distribution just by Figure 5 is a bit arbitrary. I think more data and analysis for its distribution can make the statement more convincing.

2)	Lack of implementation for LingBo. It is not clear how LingBo is implemented. Is it implemented in Linux kernel or other user-space platform such as QUIC? More detail for the implementation can make this paper stronger.

3)	In the evaluation, most of the results are about the latency and throughput. There are not evaluation for the accuracy of the RTTmin. I think more evaluation for the RTTmin can prove the effectiveness of RTTmin perception directly.

**Reviewer Confidence:**

4: The reviewer is certain that the evaluation is correct and very familiar with the relevant literature

**Scope:**

3: The work is somewhat relevant to the Web and to the track, and is of narrow interest to a sub-community

---

### Official Review · Reviewer_eZgU · 2023-11-20

**Novelty:** 4
**Technical Quality:** 5

**Review:**

### Major issues addressed in the paper:

This paper points out the problem that congestion control mechanisms, which use RTT as a metric to prevent overload, must take into account the strong jitter in wireless scenarios. For this purpose, a 50h test measurement was performed to determine the jitter of the minimum RTT in different scenarios. After describing the problem, a method (LingBo) is presented based on the test traces. RTTmin is assumed to be a normally distributed variable and a neural network is trained with the network parameters (offline). Online, the trained network is used to predict the congestion window based on the network characteristics. The procedure is evaluated intensively, on the one hand with emulated values, but also in a real scenario. Overall, it shows good performance in terms of throughput and delay, with LingBo responding in a balanced way in all scenarios (wired, wifi cellular).

### Detailed comments:

Basically, it is a solid paper, but in general I have to say that there is a bit too much photos in some places. For example in the motivation (page 1,row 73,ff). The knowledge that the jitter of the RTTmin is significantly higher in a wifi/cellular network is not a new finding. Or page 6 (row 633) "... the only algorithm that performs well"... It is not wrong that jitter is a problem and that it has been proven by traces. LingBo also performs quite well, but not outstanding compared to others.

The paper is easy to read and the approaches are easy to follow. Ultimately, the approach is also straight forward. In particular because LingBo relies on pre-training, a clearer differentiation from other methods would be desirable. What exactly is the difference to other methods such as BBR? Related work and general mechanisms are discussed, but the direct differentiation, or whether LingBo is merely a slightly better tuned variant, remains unclear.

The evaluation against a large number of other CC algorithms is good. I'm not that deep into the subject and can't estimate the actual effort involved, but in my opinion the evaluation is very comprehensive and sheds light on the relevant aspects of throughput, delay, different procedures, packet error rates, buffer sizes, etc.

In Figure 10 (a,b,c) it is noticeable that LingBo has a very high variance in relation to the delay. The average performance, both in throughput and delay, is quite good compared to the other methods (in all scenarios). However, the high variance is a disadvantage that should be discussed.
It is also striking in this context that this high variance does not occur in the evaluation in the real scenario.

Overall, the scenarios remain somewhat unclear. Measurements were taken from different locations to different locations. Wired/wifi/cellular scenarios were considered. Beyond that, there is virtually no information. How was the general network utilization, especially with wifi the jitter is caused by the media access, but no information on this. It is therefore a pity that Figure 1 only shows a section of the data.
For the evaluation in the real setting, it would also have been good to indicate the actual jitter (Figure 13). As already mentioned, LingBo has hardly any variance here. Why?

Finally, it would have been good to determine the computational overhead. The effort involved in adapting to a new scenario, since the pre-trained phase depends on the network characteristics, it is questionable how well the transferability to dynamic scenarios is given at all.

**Questions:**

What was the general network utilization in WiFi/cellular scenarios? Why were only exemplary jitter values presented?

What particular difference is there to other methods that use RTT metrics from the network characteristics, e.g. BBR?

How good is the adaptability compared to other network scenarios? What training time does the method require to adapt (e.g. in dynamic scenarios)?

What overhead (computational) does the method generate?

Why is there such a high variance in delay in the emulated scenario with LingBo, while there is hardly any in the real scenario?

**Reviewer Confidence:**

2: The reviewer is willing to defend the evaluation, but it is likely that the reviewer did not understand parts of the paper

**Scope:**

3: The work is somewhat relevant to the Web and to the track, and is of narrow interest to a sub-community

---

### Official Review · Reviewer_HExN · 2023-11-23

**Novelty:** 1
**Technical Quality:** 3

**Review:**

This paper presents a ML-trained congestion control system, targeting jitter in wireless networks. The paper is called "Meet challenges of RTT Jitter, A Hybrid Internet Congestion Control Algorithm", but is actually not describing an algorithm, but rather the structure of the ML system.

## Assessment

There are many ML-based congestion control systems. When you train your model with data from specific networks (here: wireless), then it's not surprising that the model would work well in such scenarios. That does not make the contribution useable for the Internet, or the Web.

**Questions:**

1. Can you explain how your contribution fits to the topic of the conference? It seems to be solely focuses on ML-based congestion control. Your evaluation does not assume web traffic?
2. Can you demonstrate how your system would work in other Internet scenarios? How implementable/usable is it on end systems?

**Ethics Review Description:**

no issues

**Reviewer Confidence:**

3: The reviewer is confident but not certain that the evaluation is correct

**Scope:**

2: The connection to the Web is incidental, e.g., use of Web data or API

---

### Official Review · Reviewer_F8eG · 2023-11-23

**Novelty:** 6
**Technical Quality:** 6

**Review:**

This paper proposes a novel congestionwe control algorithm, named LingBo, which leverages a decision model to predict the  RTTmin distribution under different network environments. Comments are provided per section;

Section 1 - Introduction
This part is written very nicely. The motivation behind LingBo is clear and the authors provide a solid overview of the methodology followed in the paper as well as a summary of their contributions.
- Potential questions: Power95 should be briefly defined early in the paper. Unless a reader is familiar with the term, it is not easy to understand this metric.

Section 2 - Related Work:
The literature review, even quite extensive, it is quite short (given the length of the paper). If you have space, it would be nice to elaborate a bit more on some of the existing papers. In addition, a paragraph to show how your work would improve upon the state-of-the-art would improve this section significantly.

Section 3 - RTTmin Jitter in Network
- y-axis scale on Figure 1 (a-c) differs across the three environments, which makes comparison harder. Same for the x-axis on Figure 1 (d-f)

Section 4 - LINGBO
- Some definition of imitation learning is required
- The description of LINGBO is overall very detailed and comprehensible

Section 5 - Evaluation
- The authors use a large pool of algorithms as a baseline along. A brief description along with a reference is provided for each algorithm.
- Figure 10: Given that the legend in each plot is identical, consider keeping a single legend among all plots, so that points are visible. Currently, some of the results are hidden behind the legend (Similar to Figure 9)
- 5.2.
1. 'excellent performance in terms of throughput and latency at 12mbps' - What is the latency here?
2. 'Similarly, LingBo, DuGu, Copa, Auraro, and Orca exhibit outstanding performance in both throughput and latency under the FCC conditions.' - Quantify what is the outstanding performance.

Section 6 - Conclusions
- Future Work could be added here

**Questions:**

Section 3.1:
- Can you provide additional details on the experimental setup?
Section 3.2:
- For Figures 1c-e, you are showing a subset of the congestion control algorithms that you are using as baseline later on in the paper? Is there a specific reason for that selection?
- Plots are nice but a bit cluttered for their size. On the one hand it makes sense since you are comparing 16 different algorithms, on the other hand however, it is generally hard to see the values for specific algorithms.

**Reviewer Confidence:**

3: The reviewer is confident but not certain that the evaluation is correct

**Scope:**

4: The work is relevant to the Web and to the track, and is of broad interest to the community

---

### Official Review · Reviewer_P5cY · 2023-11-23

**Novelty:** 3
**Technical Quality:** 4

**Review:**

This paper proposed Lingbo, which is a congestion control algorithm designed for environments with presence of high jitter and RTTmin variance. The paper conducts a measurement study to show that RTTmin/jitter in WiFi and cellular network is high which causes RTTmin to fluctuate. It then learns a distribution of RTTmin and uses it to predict the ideal cwnd size. Evaluations are done in a simulated environment which shows that Lingbo achieves high throughput and low latency against a range of baselines.

### Pros:
- Comparison against a large number of baselines.
- Congestion control is always an important topic.
- Measurement driven work.

### Cons:
 - The paper needs writing improvements, some parts are hard to understand.
- Evaluation doesn’t provide much insights into performance gains.
- Parts of the design are not clearly explained/motivated.

**Questions:**

**Sec1:** *Unlike the exploration of maximum bandwidth, exploring RTT𝑚𝑖𝑛 can potentially lead to a loss in bandwidth utilization.*
What is meant by RTTmin exploration and does it lead to loss in bandwidth utilization? Can the paper explain this better.

**Sec 1:** *achieves the highest power95*. Is there a particular reason why this metric is chosen instead of just throughput and latency?

## Design
- How is the interval of 100msec chosen to update the congestion window size?

- Is there any validation performed on the simulation setup to determine if it faithfully captures real world scenarios?

- It is unclear if future RTTmin is learnable/predictable with high accuracy, is there any underlying pattern that makes the problem amenable to learning? In other words, if we adapted cwnd by simply drawing a sample from a uniform or normal distribution of RTTmin, would the data-learnt approach out perform it?

- Sec 4.6 uses decision models which are trained offline. How are these models trained, can the paper provide more details here?

## Evaluation
- How is the dataset split between test and training? And how do we know that Lingo is not overfitting?

- While the results are positive, the evaluation doesn’t demonstrate Lingbo’s accuracy in predicting RTTmin, as such it is unclear whether the achieved gains are due to some other underlying phenomenon. The evaluation needs to separately dissect Lingbo’s design.

- What gives Lingbo performance advantage in Wired networks (**Sec 5.3**) where the RTT remains stable? The paper needs to provide deeper insights rather than a commentary on the results.

**Reviewer Confidence:**

2: The reviewer is willing to defend the evaluation, but it is likely that the reviewer did not understand parts of the paper

**Scope:**

3: The work is somewhat relevant to the Web and to the track, and is of narrow interest to a sub-community

---

### Decision · Program_Chairs · 2024-01-22

**Decision:**

Accept

**Comment:**

The paper introduces LingBo, a TCP mechanism designed to address the instability of RTTmin, demonstrated through real network traces. Strengths include being the first to validate RTTmin instability with real data and proposing improvements to the RTTmin perception phase. The reviewers mention that the evaluation is solid, and the authors compare LingBo with various TCP mechanisms. However, reviewers mentioned weaknesses include a lack of demonstration for the effectiveness of RTTmin perception and insufficient details on LingBo's implementation. Questions raised concern the assumption of RTTmin distribution, the absence of RTTmin accuracy evaluation, and clarity on LingBo's implementation.

On reviewer in particular was very negative regarding this work, but did not provide sufficient information about the reasons to reject this work, not did he engage further with the authors during rebuttal phase.

Overall, I encourage the authors to address comments such as the lack of clarity in the implementation details, insufficient demonstration of the proposed solution's effectiveness, and questions about assumptions made in the paper. Despite these issues, the paper shows promise, and the positive aspects, such as the novel observation and robust evaluation, could generate interesting discussions.